# Simultaneous Determination of Seven Lipophilic and Hydrophilic Components in *Salvia miltiorrhiza* Bunge by LC-MS/MS Method and Its Application to a Transport Study in a Blood-Brain-Barrier Cell Model

**DOI:** 10.3390/molecules27030657

**Published:** 2022-01-20

**Authors:** Hui Wang, Mingyong Zhang, Jiahao Fang, Yuzhen He, Min Liu, Zhanying Hong, Yifeng Chai

**Affiliations:** 1School of Pharmacy, Naval Medical University, Shanghai 200433, China; wanghuiuuh@163.com (H.W.); myzhangchina@163.com (M.Z.); fangjiahaos@163.com (J.F.); heyuzhen9477@163.com (Y.H.); yfchai@smmu.edu.cn (Y.C.); 2Shanghai Key Laboratory for Pharmaceutical Metabolite Research, Shanghai 200433, China; 3Department of Pharmacy, Changhai Hospital, Naval Medical University, Shanghai 200433, China; lm_yaofen2003@163.com

**Keywords:** *Salvia miltiorrhiza* Bunge, LC-MS/MS, active components, transmembrane transport, BBB cell model

## Abstract

*Salvia miltiorrhiza* Bunge (SM) has been extensively used in Alzheimer’s disease treatment, the permeability through the blood-brain barrier (BBB) determining its efficacy. However, the transport mechanism of SM components across the BBB remains to be clarified. A simple, precise, and sensitive method using LC-MS/MS was developed for simultaneous quantification of tanshinone I (TS I), dihydrotanshinone I (DTS I), tanshinone IIA (TS IIA), cryptotanshinone (CTS), protocatechuic aldehyde (PAL), protocatechuic acid (PCTA), and caffeic acid (CFA) in transport samples. The analytes were separated on a C18 column by gradient elution. Multiple reaction monitoring mode via electrospray ionization source was used to quantify the analytes in positive mode for TS I, DTS I, TS IIA, CTS, and negative mode for PAL, PCTA, and CFA. The linearity ranges were 0.1–8 ng/mL for TS I and DTS I, 0.2–8 ng/mL for TS IIA, 1–80 ng/mL for CTS, 20–800 ng/mL for PAL and CFA, and 10–4000 ng/mL for PCTA. The developed method was accurate and precise for the compounds. The relative matrix effect was less than 15%, and the analytes were stable for analysis. The established method was successfully applied for transport experiments on a BBB cell model to evaluate the apparent permeability of the seven components.

## 1. Introduction

Alzheimer’s disease (AD) is the leading cause of dementia and one of the significant healthcare challenges of the 21st century [1]. AD is an age-related neurodegenerative disorder characterized by progressive cognitive decline and memory loss [2]. Research since the discoveries of amyloid β (Aβ) and tau protein, the main components of senile plaque (SP), and neuro-fibrillary tangles (NFT), respectively, has provided detailed information about molecular pathogenetic events of AD [3]. Additionally, oxidative stress, mitochondrial dysfunction, excessive reactive oxygen species production, lipid peroxidation, proteasomal dysfunction, microglial activation, neurotransmitter alteration, and neuroinflammation have also been implicated in AD pathology [4]. However, the cause of AD is poorly known, and there are no curative treatments despite the major expenditure of research and money over many decades [5]. Current treatment strategies only provide symptomatic relief. The common drugs, including cholinesterase inhibitors, N-methyl-D-aspartic acid (NMDA) receptor antagonists, and anti-inflammatory and antioxidant agents, have various adverse effects [6]. In China, traditional Chinese medicine (TCM) has a long history of treatment for AD, and extensive progress research has been conducted for the prevention and treatment of AD [7].

*Salvia miltiorrhiza* Bunge (SM), also known as Danshen in Chinese, has been extensively used in clinics to treat stroke perimenopausal syndrome, anemia, and cardiovascular and cerebrovascular diseases for hundreds of years in China [8]. Increasing studies demonstrated that SM significantly improved the symptoms of AD and other central nervous system diseases [9,10]. However, the transport mechanism of SM across the blood-brain barrier (BBB) remains to be clarified. The BBB plays a vital role in maintaining the balance and stability of the brain microenvironment by maintaining restricted transport of toxic or nutrimental molecules and the removal of metabolites [11]. The BBB constitutes multiple physical and chemical barriers that restrict the movement of drugs and antigens across it, leading to minimal bioavailability of drugs in the central nervous system (CNS) [12]. As the transport mechanism of SM across the BBB has not been fully elucidated, it is necessary to establish a method to determine active components of SM for the study of transport across the BBB [13].

According to their structural characteristics, the active ingredients of SM can be classified into two groups: hydrophilic phenolic acids, such as salvianolic acid B (Sal B), salvianolic acid A (Sal A), and danshensu, protocatechuic aldehyde (PAL), protocatechuic acid (PCTA), caffeic acid (CFA), and lipophilic tanshinones, such as tanshinone I (TS I), dihydrotanshinone (DTS), tanshinone IIA (TS IIA), and cryptotanshinone (CTS) [14]. Studies have shown that both groups have multiple neuroprotective potentials relevant to AD, such as anti-Aβ, antioxidant, anti-apoptotic, and anti-inflammation properties, and enhancement of cholinergic transmission [15]. High-performance liquid chromatography tandem triple quadrupole mass spectrometry has been widely used in the quantification of active components in SM [16,17,18]. An ultra-performance liquid chromatography tandem mass spectrometry analysis method was established and detected 12 phenolic acids and five tanshinones in SM extract solutions [16]. Liu et al. [18] established an LC-MS method for separating and detecting RA, TS I, CTS, TS IIA, and DTS I in plasma of rats after oral administration of SM. However, the samples used were plasma and brain tissue, and no research has been paid to simultaneous determination of SM components in transport samples.

This study aimed to develop an LC-MS/MS method to simultaneously determine four lipophilic tanshinones, TS I, DTS I, TS IIA, CTS, and three hydrophilic phenolic acids, PAL, PCTA, and CFA, in HBSS (Hank’s Balanced Salt Solution) samples (as shown in Figure 1). Then, the LC-MS/MS method was applied for transmembrane transport study of these active components in SM on a BBB cell model. 

## 2. Results and Discussion

### 2.1. Method Development

To achieve a rapid and reliable LC-MS/MS method for determining ingredients of SM in HBSS, optimization of chromatographic separation and mass spectrometric detection parameters was systematically carried out. Seven ingredients with good stability in HBSS were selected, including water-soluble compounds PAL, PCTA, CFA, and lipid-soluble components TS IIA, CTS, TS I, and DTS I, to establish a sensitive LC-MS/MS method. In subsequent experiments, the results demonstrated that the water-soluble analytes in the negative ionization mode expressed high intensity and good sensitivity of precursor and product ions. In contrast, the lipid-soluble analytes responded well in the positive ionization mode. 

For the chromatographic separation, physicochemical properties demonstrated that the seven target compounds possessed an extensive polarity range. Therefore, the gradient elution method was used to achieve high separation efficiency on the C18 column. Formic acid in water (solvent A) and acetonitrile (solvent B) were selected after optimization. A total of 0.05% formic acid was added to enhance the mass spectrometry ionization and maintain a good peak shape. Afterward, the influence of different flow rates (0.300 mL/min, 0.400 mL/min) was experimentally tested. The results showed no significant differences in the retention time among these conditions, while the column pressure at 0.400 mL/min was increased. Thus, 0.300 mL/min was chosen as the final flow rate considering the time and cost saving. Moreover, the applicability of sulfamethoxazole (SMZ) and simvastatin was investigated as internal standards (IS). The internal standard SMZ had a good separation with the analytes and showed intense signal responses and less noise in the negative mode. At the same time, simvastatin, as a lactone compound, was very unstable in the HBSS system. 

In the method described, the mobile phase consisted of water and acetonitrile using gradient elution. In the first 8 min, all of the water-soluble analytes in the negative ionization mode, including the internal standard, were washed out in the negative mode. The lipid-soluble compounds in the positive mode were eluted after 8 min. Therefore, ESI- scanning was used in the first 8.5 min, and ESI + scanning was used after 8.5 min to avoid reducing the instrument’s sensitivity due to frequent switching of positive and negative ion modes. The HBSS balanced salt, as the transport system, contains many inorganic salt ions. To protect the mass spectrometer from HBSS, the divert valve was set to direct the flow to the waste from 0 to 1.7 min. The highlight of this study is simultaneous quantification of the seven compounds in both positive and negative ion modes with the same internal standard. 

### 2.2. Method Validation

Comprehensive method validation was performed in specificity, linearity and range, accuracy and precision, matrix effect, and stability.

#### 2.2.1. Specificity

Specificity was assessed by comparing the LC-MS/MS chromatogram integrity of the blank HBSS and the lowest limit of quantification (LLOQ) samples spiked with IS and the transport sample. The developed method showed no significant interfering peak from endogenous substances interferences at the retention times of seven SM components and IS in HBSS and transport samples (as shown in Figure 2). CTS had a similar retention time as the TS I, but they do not share the same fragmentation ion at the multiple reaction monitoring (MRM) mode. With the analysis of the purity of the mass chromatogram, this method can be used for the determination of seven components of SM in HBSS and transport samples. 

#### 2.2.2. Linearity and LLOQ

Calibration standards (CS), with a concentration range of 0.1–8 ng/mL for TS I and DTS I, 0.2–8 ng/mL for TS IIA, 1–80 ng/mL for CTS, 20–800 ng/mL for PAL and CFA, and 10–4000 ng/mL for PCTA, were prepared and analyzed in duplicate in three separate analytical runs. Linear regression analysis was performed and correlation coefficients (r^2^) for calibration generated were 0.98 for CTS, and were greater than 0.99 for other six components in HBSS, indicating good linearity over the range studied for the seven components of SM. The regression equations, correlation coefficients, and linear ranges of seven components of SM are shown in Table 1.

#### 2.2.3. Precision and Accuracy

The intra-day precision and accuracy were evaluated by analyzing five replicate QC samples at three levels, low QC (LQC), medium QC (MQC), and high QC (HQC). The inter-day precision and accuracy were determined by running three validation batches on each of three consecutive days. As shown in Table 2, the intra-day and inter-day precision and accuracy of seven components at high, medium, and low concentrations were within the acceptable range, except for the intra-day accuracy of CFA at high concentrations. The CFA’s intra-day accuracy of the highest concentration on the first day was 81.51%, but its intra-day accuracy was within the acceptable range. For the six components of SM except for CFA, the precision of LOQ of each component was not exceeding 20%, and the accuracy of LOQ of each compound was within 80–120%. Therefore, this method was considered to be accurate and precise.

#### 2.2.4. Matrix Effect

Matrix effects of seven compounds of SM at low, medium, and high QC are shown in Table 3, respectively. The matrix effect of the target compound was similar at LQC, MQC, and HQC concentrations, all of which were enhanced or inhibited. Furthermore, for seven compounds, the RSD of the internal standard normalized matrix factor from the six batches appeared to be less than 15% at both tested concentration levels, thus meeting the acceptance criteria. Therefore, seven compounds of SM can be accurately quantified in positive and negative ion modes by the same IS in this research.

#### 2.2.5. Stability

Although the samples were processed immediately, stability of the analytes at autosampler for 24 h and at −80 °C for 15 days was also assessed in the current study to extend the application. As shown in Table 4, the RSD of stability of each component was less than 15%, indicating good stability of seven compounds in HBSS. The good stabilities of the components ensured the veracity of the quantitation results by this method.

### 2.3. BBB Cell Model Transport Study Application

The validated LC-MS/MS method was applied to evaluate the permeability of the seven components of SM in a BBB cell model. The trans-epithelial electric resistance (TEER) reached the maximum value of 33.04 ± 5.02 Ω·cm^2^ two days after hBMEC cells were seeded on the apical side of the transwell, which is close to the reported value, and meets the requirements of the in vitro BBB model [19]. By measuring the concentration of SM components in the samples obtained from the transport experiment, the apparent permeability coefficient (P_app_) and efflux rate (ER) cumulative transport was calculated, and the results are shown in Table 5. On the BBB cell model, within 120 min, the P_app (AP-BL)_ of CFA, PAL, and PCTA was greater than 1 × 10^−6^ cm/s, indicating that they were easily absorbed in the BBB model. The P_app (AP-BL)_ of DTS I and CTS were greater than 1 × 10^−7^ cm/s and less than 1 × 10^−6^ cm/s, indicating moderate absorption in the model. TS IIA’s P_app (AP-BL)_ was less than 1 × 10^−7^ cm/s, indicating it was difficult to be absorbed in the model [20]. Generally speaking, when the ER is greater than 2.0, it may be considered a substrate of the efflux transporter, and when the ER is less than 0.5, the compound is actively taken up [21,22]. 

The ER values of TS IIA, DTS I, CFA, CTS, PAL, and PCTA were 0.673, 0.779, 1.049, 1.030, 0.805, and 0.946, respectively, indicating that the absorption of the above compounds was close to their efflux rates in the BBB cell model. These results suggest that the transport mode of these compounds in the BBB cell model established in this study may be passive diffusion [23]. Treatment with TS IIA was found to suppress disruption of the BBB [24], and also polarized transport of CTS was found with facilitated efflux from the abluminal side to luminal side in endothelial cell monolayers [25], which are consistent with our results and would be scientific evidence to support the application of the components of SM to a certain extent.

## 3. Materials and Methods

### 3.1. Chemicals

TS I, DTS I, TS IIA, CTS, PAL, PCTA, and CFA reference standards were purchased from Shanghai Standard Technology Co., Ltd. (Shanghai, China). SMZ, used as the internal standard, was supplied by the National Institutes for Food and Drug Control (Beijing, China). Hank’s Balanced Salt Solution (HBSS) was supplied by Gibco (Thermo Scientific, Waltham, MA, USA). Acetonitrile, methanol, and formic acid (HPLC grade) were purchased from CNW (Shanghai, China). Dimethyl sulfoxide (DMSO) was supplied by Thermo Scientific (Waltham, MA, USA). Water was prepared from the Merck ultra-pure water apparatus. Other reagents were of analytical grade.

### 3.2. Cell Culture

Endothelial cell medium (ECM), penicillin, fetal calf serum, trypsin, streptomycin, HBSS, and other culture reagents were purchased from Gibco (Carlsbad, CA, USA). Human brain microvascular endothelial cells (hBMECs) were purchased from ScienCell (Carlsbad, CA, USA). 12-well transwell plates with polystyrene inserts (0.4 μm pore size and 12 mm in diameter) were obtained from Corning Costar (Cambridge, MA, USA). The epithelial voltammeter was obtained from Electrical Resistance System (Millicell ERS-2, Canton, MA, USA). hBMECs cells were maintained in ECM supplemented with 10% fetal bovine serum (FBS),100 units/mL penicillin and 100 mg/mL streptomycin in 5% CO_2_ at 37 °C.

### 3.3. Instrumentation and Conditions

Quantitative analysis of the seven components was performed by liquid chromatography-triple quadrupole tandem mass spectrometry (LC-MS/MS), which consisted of an HPLC system (Agilent 1260 Infinity, Santa Clara, CA, USA) coupled with a triple quadrupole mass spectrometer with an electrospray ionization interface (Agilent 6460, Santa Clara, CA, USA).

Chromatographic separation was performed using an Agilent poroshell 120 EC-C18 column (3.0 mm × 100 mm, 2.7 μm) maintained at 25 °C. The flow rate was 0.3 mL/min, and the volume of injection was 5 μL. The mobile phase consisted of 0.05% formic acid in an aqueous solution as solvent A and acetonitrile solution as solvent B. The column was eluted with a gradient of 85–50% A at 0–3.5 min, 50–10% A at 3.5–6 min, 10–5% A at 6–7.5 min, and 5% A at 7.5–14 min.

Mass spectrometry quantitative detection was operated in positive mode and negative mode. The MS parameters were as follows: capillary voltage of 4.0 kV (ESI+)/3.5 kV (ESI−), fragmentor voltage of 120.0 V, drying gas flow of 11.0 L/min, and a gas temperature of 350 °C. The MRM mode was employed to quantify PAL, PCTA, CFA, and SMZ in negative ion mode in the first 8.5 min, and TS I, TS IIA, DTS I, and CTS in positive ion mode after 8.5 min. The related parameters were listed in Table 6. 

### 3.4. Stock Solutions, Calibration Solutions, and Quality Control Samples

Stock solutions of the TS I, TS IIA, DTS I, and CTS were prepared by dissolving each accurately weighed standard in DMSO and further diluting with methanol to obtain a 10 μg/mL final concentration. Stock solutions of the PAL, PCTA, CFA, and SMZ (IS) were prepared by dissolving each accurately weighed standard in methanol to obtain a 1 mg/mL final concentration.

The mixed working solution for calibration standard containing 100 ng/mL of TS I, DTS I and TS IIA, 1000 ng/mL of CTS, 10 μg/mL of PAL and CFA, and 50 μg/mL PCTA was prepared by diluting the stock solutions in methanol, and further diluted to a series of concentrations with methanol. Working solutions for QC samples at three levels were obtained using different preparations of stock solutions by diluting analytes in methanol. The concentrations of seven reference compounds in QC working solutions were as follows: 2, 10, and 80 ng/mL for TS I; 2, 10, and 80 ng/mL for DTS I; 4, 20, and 80 ng/mL for TS IIA; 20, 200, and 800 ng/mL for CTS; 400, 2000, and 8000 ng/mL for PAL; 200, 2000, and 4000 ng/mL for PCTA; and 400, 2000, and 8000 ng/mL for CFA. The internal standard (IS) working solution was obtained by diluting the IS stock solutions in methanol, yielding a 200 ng/mL solution. All working solutions for CS and QC were stored at −80 °C. CS samples and QC samples were prepared by mixing 10 μL of working solution with 90 μL of HBSS, respectively.

### 3.5. Sample Preparation

A 20 μL of IS working solution was added to 100 μL of CS or QC samples, and samples were mixed and centrifuged at 4 °C, 12,000 rpm for 15 min. The supernatant was transferred to autosampler vials with inserts before analysis. A volume of 5 μL of the supernatant was injected into the LC-MS/MS system.

### 3.6. Validation Procedure

A total of six different batches of blanks were spiked at the LLOQ level and were processed to assess the selectivity of the assay. LC-MS/MS chromatograms of the blanks and LLOQ samples and the transport samples were monitored and compared for chromatographic integrity and potential interferences. The seven analytes in spiked samples should be identifiable, discrete, and reproducible.

Linearity and range were investigated by constructing a calibration curve based on a series of concentrations. The calibration curve was constructed by plotting the peak area ratios (y) of analytes to IS versus the concentrations of the analytes, using weighted linear least-squares regression. The LLOQ was defined as the concentration producing S/N ratio of 10. The calibration curves were acceptable if 75% of all non-zero CS were within ±15% of the nominal concentrations or ±20% of the lower limit of quantification. 

A total of five replicates of LQC, MQC, HQC were analyzed in three analytical runs. The accuracy and precision were expressed as relative errors (RE) and relative standard deviation (RSD), respectively. The mean value of accuracy should be within ±15% of the nominal value. The precision determined at each concentration level should be an RSD less than 15%.

The matrix effect was evaluated by comparing peak areas of spiked samples (the peak area in HBSS) to the mixed standard solution (the peak area in a neat solution) at high, medium, and low concentrations. The internal standard normalized matrix effect was calculated. 

Stability was evaluated by analyzing LQC, MQC, and HQC concentrations at autosampler for 24 h and −80 °C for 15 days with five replicates. Stability was expressed in terms of accuracy (RE) and coefficient of variation (RSD). The samples were considered stable if accuracy (RE) was within ±15% of the nominal values and precision (RSD) was less than 15%.

### 3.7. Transport Study in a BBB Cell Model

The hBMECs cells were seeded at a density of 3 × 10^5^ cells per well on the transwell plates in EMC. Apical side volumes were 0.5 mL, and the medium was changed every day. Basal side volumes were 1.5 mL, and the medium was changed every other day. The medium was changed until confluent monolayers were formed. The integrity and transport capacity of the hBMEC cell monolayer was checked by measuring the TEER using an epithelial voltammeter, following the protocols suggested by the manufacturer. TEER provides information on ion current resistance across cell monolayers related to the integrity of tight junctions between cells. When the TEER reached the maximum value, the cell monolayer was used for the transport assay. 

The permeability experiments were carried out bi-directionally, with working solutions containing one test component, TSI (1.25 μg/mL), DTSI (0.15 μg/mL), CTS (1.25 μg/mL), CFA (10 μg/mL), PAL (10 μg/mL), and PCTA (20 μg/mL). Aliquots of the donor compartment were collected after several time points (15, 30, 60, and 120 min) and replaced with drug-free assay buffer. The transwell plates were incubated on an orbital shaker at 37 °C for the entire experimental time. All experiments were performed in triplicate, and withdrawn samples were stored in −80 °C fridge before LC-MS/MS analysis. 

For the BBB cell model, the apparent permeability coefficient, P_app_ in apical-to-basolateral (AP-BL) or basolateral-to-apical (BL-AP) direction, and efflux rate of each component were subsequently calculated according to the following equation: P_app_ = {dQ/dt} × {1/ (A × C_0_)},(1)
where Q is the accumulation quantity of the compound in the receiver side (μg), dQ /dt is the rate of appearance of the compound in the receiver side (μg /s), C_0_ is the initial concentration in the donor side (μg), and A is the surface area of the membrane insert (cm^2^). Furthermore, efflux ratios (ER) were calculated according to the following equation: ER = P_app (BL-AP)_/P_app (AP-BL)_,(2)
where P_app (BL-AP)_ and P_app (AP-BL)_ are the P_app_ values in the direction basolateral-to-apical and apical-to-basolateral, respectively.

## 4. Conclusions

This study developed a simple, precise, and sensitive LC-MS/MS method to simultaneously determine TS I, DTS I, TS IIA, CTS, PAL, CFA, and PCTA in HBSS samples. A total of seven compounds in positive and negative ion modes can be accurately quantified with the same IS. The method was well validated in terms of specificity, linearity, LLOQ, accuracy, precision, matrix effect, and stability, and the results met the FDA’s requirements of drug quantitative analysis. The established method was successfully applied for transmembrane transport experiments to determine seven target components on a BBB cell model. This is the first study that simultaneously analyzed TS I, DTS I, TS IIA, CTS, PAL, CFA, and PCTA in HBSS samples. The application of transmembrane transport study on a BBB cell model revealed the absorption of the above compounds was close to their efflux rates in the BBB cell model, indicating that the transport mode of these compounds in the BBB cell model may be passive diffusion. The results provided a pre-clinical insight into the interaction between SM components and transporters on the BBB, which would be scientific evidence to support the better application of SM in treating AD to a certain extent.

## Figures and Tables

**Figure 1 molecules-27-00657-f001:**
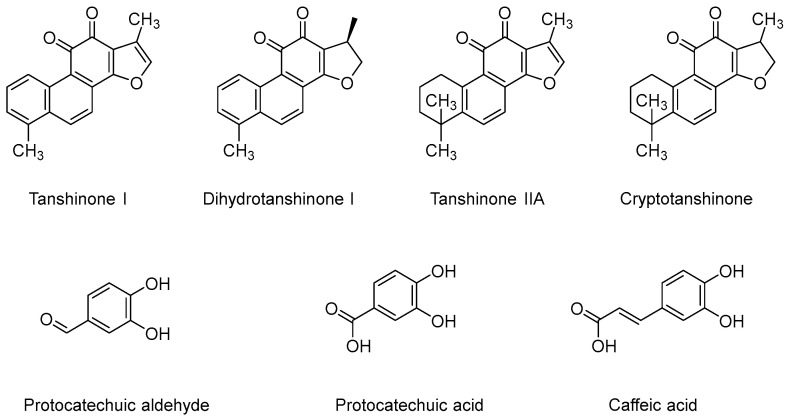
Chemical structures of seven active components in *Salvia miltiorrhiza* Bunge.

**Figure 2 molecules-27-00657-f002:**
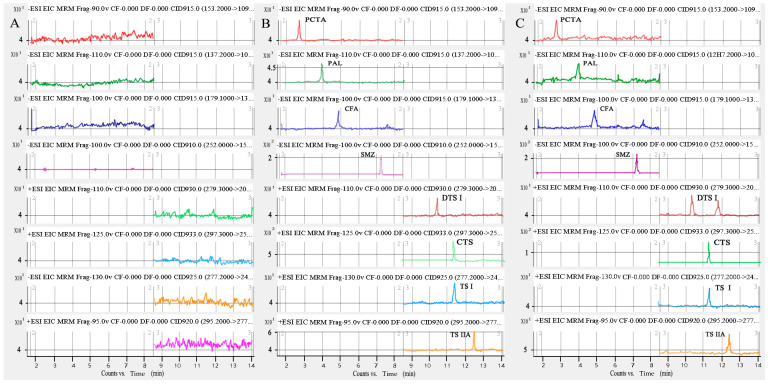
MRM chromatograms of seven components in *Salvia miltiorrhiza* Bunge and IS. (**A**) a blank HBSS sample; (**B**) a blank HBSS sample spiked with IS and seven components at LLOQ level; (**C**) The transport sample.

**Table 1 molecules-27-00657-t001:** Calibration curves of the seven components of *Salvia miltiorrhiza* Bunge in HBSS.

Analytes	Calibration Curve	Weighting	Linear Range (ng/mL)	r^2^	LLOQ (ng/mL)
TS I	Y = 0.1068X + 0.0102	1/X^2^	0.1–8	0.9934	0.1
DTS I	Y = 0.2497X + 0.0043	1/X	0.1–8	0.9988	0.1
TS IIA	Y = 0.8962X + 0.0426	1/X^2^	0.2–8	0.9922	0.2
CTS	Y = 0.2353X + 0.0537	1/X^2^	1–80	0.9837	1
PAL	Y = 0.0024X + 0.0092	1/X	20–800	0.9965	20
PCTA	Y = 0.0124X + 0.0451	1/X^2^	10–4000	0.9965	10
CFA	Y = 0.0195X + 0.0250	1/X	20–800	0.9989	20

**Table 2 molecules-27-00657-t002:** Accuracy and precision of seven constituents of *Salvia miltiorrhiza* Bunge in HBSS.

Analytes	Con. (ng/mL)	Validation Run 1			Validation Run 2			Validation Run 3			Between-Run		
		Mean ± SD (ng/mL)	Accuracy (%)	RSD (%)	Mean ± SD (ng/mL)	Accuracy (%)	RSD (%)	Mean ± SD (ng/mL)	Accuracy (%)	RSD (%)	Mean ± SD (ng/mL)	Accuracy (%)	RSD (%)
TS I	0.2	0.17 ± 0.02	86.46 ± 9.86	9.83	0.22 ± 0.01	107.58 ± 6.93	5.77	0.21 ± 0.01	104.97 ± 8.08	6.87	0.20 ± 0.03	99.67 ± 12.44	12.88
	1	1.00 ± 0.03	100.21 ± 2.91	2.59	1.13 ± 0.03	113.05 ± 2.91	2.67	1.09 ± 0.06	108.61 ± 6.61	5.44	1.07 ± 0.07	107.29 ± 6.97	6.49
	8	7.52 ± 0.24	94.06 ± 3.34	3.17	7.61 ± 0.25	95.10 ± 3.54	3.32	8.16 ± 0.56	102.06 ± 8.10	6.87	7.74 ± 0.47	96.72 ± 5.91	6.10
TS IIA	0.4	0.34 ± 0.01	85.18 ± 3.95	4.25	0.42 ± 0.01	104.73 ± 4.04	3.46	0.35 ± 0.01	87.88 ± 3.55	3.51	0.37 ± 0.04	92.93 ± 9.86	10.44
	2	1.90 ± 0.07	95.14 ± 4.08	3.82	2.13 ± 0.09	106.48 ± 5.15	4.34	2.07 ± 0.04	103.54 ± 2.12	1.93	2.03 ± 0.12	101.72 ± 6.19	6.09
	8	8.57 ± 0.17	107.08 ± 2.33	1.95	8.98 ± 0.20	112.23 ± 2.79	2.22	8.36 ± 0.70	104.47 ± 9.85	8.43	8.63 ± 0.52	107.93 ± 6.53	6.05
DTS I	0.2	0.18 ± 0.01	87.28 ± 4.63	4.45	0.20 ± 0.01	102.02 ± 5.19	4.54	0.18 ± 0.02	91.9013.54	13.16	0.19 ± 0.02	93.73 ± 10.34	10.86
	1	0.97 ± 0.03	97.16 ± 3.27	3.01	1.07 ± 0.04	107.054.69	3.93	0.93 ± 0.04	93.10 ± 4.91	4.72	0.99 ± 0.07	99.10 ± 7.28	7.35
	8	7.14 ± 0.21	89.29 ± 2.88	2.88	7.91 ± 0.93	98.89 ± 13.01	11.75	6.81 ± 0.51	85.17 ± 7.11	7.47	7.29 ± 0.80	91.12 ± 10.03	11.00
CTS	2	1.83 ± 0.05	91.60 ± 2.26	2.46	2.03 ± 0.06	101.61 ± 3.48	3.06	2.23 ± 0.05	111.61 ± 3.18	2.45	2.03 ± 0.17	100.89 ± 8.74	8.54
	20	17.73 ± 0.32	88.67 ± 1.77	1.80	19.00 ± 0.67	94.99 ± 3.75	3.53	22.26 ± 0.54	111.30 ± 3.04	2.44	19.66 ± 2.05	98.32 ± 10.24	10.43
	80	69.54 ± 1.78	86.92 ± 2.49	2.55	71.83 ± 1.22	89.78 ± 1.77	1.70	70.83 ± 1.39	88.53 ± 1.97	1.96	70.65 ± 1.83	99.31 ± 2.29	2.58
PAL	40	39.61 ± 6.39	99.01 ± 3.53	3.09	40.43 ± 1.56	101.09 ± 4.53	3.87	45.40 ± 1.28	113.50 ± 3.58	2.82	42.09 ± 3.09	105.22 ± 7.72	7.33
	200	215.23 ± 6.35	107.61 ± 3.57	2.95	217.97 ± 4.90	108.99 ± 2.75	2.25	224.32 ± 3.71	113.15 ± 2.75	1.65	219.18 ± 6.59	109.58 ± 3.30	3.01
	800	811.61 ± 12.22	101.46 ± 1.71	1.51	808.07 ± 4.92	101.02 ± 0.70	0.61	778.96 ± 53.75	98.63 ± 7.73	6.81	803.87 ± 32.47	100.49 ± 1.05	4.04
PCTA	20	22.87 ± 0.76	113.70 ± 3.79	3.31	18.30 ± 1.35	91.43 ± 4.78	7.37	18.34 ± 0.57	91.70 ± 3.18	3.12	19.62 ± 2.36	97.89 ± 11.01	12.01
	200	228.23 ± 9.87	109.51 ± 4.96	1.26	204.31 ± 2.58	102.16 ± 1.44	1.26	206.55 ± 7.03	103.30 ± 3.93	3.40	213.03 ± 13.40	104.99 ± 4.82	6.29
	4000	3710.42 ± 288.72	92.76 ± 8.07	7.78	3556.74 ± 44.97	88.91 ± 1.25	1.26	3688.29 ± 34.90	92.20 ± 0.99	0.95	3651.82 ± 189.36	91.29 ± 4.74	5.19
CFA	40	37.74 ± 1.59	94.66 ± 4.04	4.22	39.00 ± 0.78	97.48 ± 2.19	2.01	39.63 ± 2.31	99.07 ± 6.45	5.83	38.87 ± 1.92	97.07 ± 4.64	4.94
	200	194.22 ± 6.35	97.12 ± 3.53	3.27	201.53 ± 13.94	100.78 ± 7.81	6.92	193.81 ± 10.40	96.90 ± 5.83	5.37	196.52 ± 11.66	98.27 ± 5.84	5.93
	800	652.11 ± 4.53	81.51 ± 0.64	0.69	728.80 ± 30.19	91.10 ± 4.37	4.14	695.55 ± 5.20	86.92 ± 0.78	0.75	688.53 ± 39.42	86.06 ± 4.93	5.73

**Table 3 molecules-27-00657-t003:** Matrix effect of seven constituents of *Salvia miltiorrhiza* Bunge at three different concentrations in HBSS.

Analytes	LQC		MQC		HQC	
	Mean ± SD (%)	RSD (%)	Mean ± SD (%)	RSD (%)	Mean ± SD (%)	RSD (%)
TS I	94.33 ± 4.46	4.73	99.28 ± 2.78	2.8	95.33 ± 1.86	1.95
DTS I	87.76 ± 3.89	4.43	88.34 ± 5.36	6.07	86.91 ± 3.96	4.55
TS IIA	86.30 ± 5.53	6.41	96.86 ± 6.81	7.03	89.81 ± 6.29	7.01
CTS	88.11 ± 7.13	8.09	91.26 ± 8.03	8.8	86.45 ± 5.86	6.78
PAL	106.06 ± 6.03	5.69	116.50 ± 14.41	12.37	118.96 ± 10.00	8.41
PCTA	101.00 ± 1.00	0.99	108.48 ± 14.50	13.36	106.06 ± 5.76	5.43
CFA	99.97 ± 9.51	9.51	103.06 ± 6.82	6.61	104.63 ± 6.77	6.47

**Table 4 molecules-27-00657-t004:** Stability of seven constituents of *Salvia miltiorrhiza* Bunge at three different concentrations in HBSS.

Analytes	Concentration (ng/mL)	Autosampler (24 h)		Long Term (−80 °C, 15 Days)	
		Mean ± SD (%)	RSD (%)	Mean ± SD (%)	RSD (%)
TS I	0.2	104.83 ± 12.38	11.81	112.64 ± 2.51	2.23
	1	94.45 ± 5.86	6.2	111.60 ± 4.40	3.94
	8	93.83 ± 3.75	4	110.22 ± 9.21	8.36
DTS I	0.2	88.70 ± 3.67	4.14	113.43 ± 10.52	9.27
	1	91.52 ± 3.06	3.34	114.08 ± 2.21	1.94
	8	89.13 ± 2.15	2.41	99.99 ± 6.54	6.54
TS IIA	0.4	100.64 ± 14.49	14.39	108.42 ± 15.37	14.17
	2	99.36 ± 13.77	13.86	113.91 ± 1.68	1.47
	8	93.08 ± 11.14	11.97	109.50 ± 5.61	5.12
CTS	2	107.58 ± 4.68	4.35	110.59 ± 4.75	4.3
	20	94.78 ± 3.77	3.97	110.62 ± 4.01	3.63
	80	94.90 ± 3.73	3.93	91.81 ± 5.50	5.99
PAL	40	88.42 ± 5.31	6.01	106.24 ± 5.99	5.64
	200	102.92 ± 11.45	11.12	101.76 ± 8.94	8.78
	800	101.06 ± 1.83	1.81	92.28 ± 4.85	5.25
PCTA	20	96.14 ± 4.10	4.27	114.06 ± 2.31	2.03
	200	102.10 ± 6.02	5.9	110.63 ± 1.60	1.45
	4000	91.74 ± 2.51	2.74	91.53 ± 5.04	5.5
CFA	40	85.7 ± 0.42	0.5	113.78 ± 1.90	1.67
	200	91.08 ± 8.57	9.41	111.44 ± 1.95	1.75
	800	91.58 ± 6.17	6.74	106.34 ± 6.10	5.74

**Table 5 molecules-27-00657-t005:** Apparent permeability coefficient and efflux ratio of *Salvia miltiorrhiza* Bunge components in BBB cell model (Mean ± SD, *n* = 3).

Analytes	P_app (AP-BL)_ cm/s	P_app (BL-AP)_ cm/s	ER
TS IIA	(3.757 ± 1.723) × 10^−8^	(2.528 ± 0.773) × 10^−8^	0.673
DTS I	(4.643 ± 2.012) × 10^−6^	(3.617 ± 1.082) × 10^−6^	0.779
CFA	(2.147 ± 1.010) × 10^−5^	(2.252 ± 0.954) × 10^−5^	1.049
CTS	(4.977 ± 1.587) × 10^−6^	(5.125 ± 1.584) × 10^−6^	1.030
PAL	(1.516 ± 0.179) × 10^−5^	(1.220 ± 0.021) × 10^−5^	0.805
PCTA	(4.369 ± 1.410) × 10^−5^	(4.132 ± 0.288) × 10^−5^	0.946

**Table 6 molecules-27-00657-t006:** The optimized mass spectrometric parameters of seven components in *Salvia miltiorrhiza* Bunge and IS.

Analytes	Precursor (*m*/*z*)	Product (*m*/*z*)	Frag. (V)	CE (eV)	Dwell	Cell Accelerator Voltage	Polarity
TS I	277.2	249.2	130	25	100	8	Positive
DTS I	279.3	204.9	110	30	100	7	Positive
TS IIA	295.2	277.2	95	20	100	7	Positive
CTS	297.3	251.4	125	33	100	7	Positive
PAL	137.2	108.2	110	30	100	7	Negative
PCTA	153.2	109.1	90	15	100	5	Negative
CFA	179.1	135.1	100	15	100	7	Negative
SMZ (IS)	252.0	156.1	100	10	100	5	Negative

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
