# Peer review of "Simultaneous Determination of Seven Lipophilic and Hydrophilic Components in Salvia miltiorrhiza Bunge by LC-MS/MS Method and Its Application to a Transport Study in a Blood-Brain-Barrier Cell Model"

_molecules, 2022, doi:10.3390/molecules27030657_

Round 1

Reviewer 1 Report

The following are some minor comments, rather of an editorial nature. In the opinion of the reviewer, the methodological description and justification were conceptually presented in this manuscript, in general, in a professional manner.

  1. Introduction
  2. There are some gaps in the text in the space between the name and the abbreviation in square brackets, which require correction in some places.
  3. Line: 64-67: in these sentences, I would change the word “anti-apoptosis” on “anti-apoptotic”, and after "enhancement of cholinergic" I would add "action" or "transmission", or something like that.
  4. LIne: 75-78: maybe in marking the research goal it is worth mentioning, signaling in what kind of matrix, in what environment such determination was developed? (I am talking about this only because earlier it was mentioned about analytical research in biological material in vivo). But this is only a suggestion for you to consider.

  1. Materials and Methods

3.1. Chemicals

  1. If possible, it would be good to indicate the starting (stock) concentrations of the reagents used or their form (especially that in section 3.4. the authors mention about starting concentrations of CS). But this is only a suggestion to consider.

3.3. Instrumentation and Conditions

  1. Line: 224-225: maybe it would be better to put these percentages right next to the numerical values and precede the dash (i.e.: “-“) with a space (something like this: “40% - 50%”)? It seems that the perception of these numerical values would look a bit optically better.

3.4. Stock Solutions, Calibration Solutions and Quality Control Samples

  1. Line: 236: there is no space after the abbreviation SMZ.

3.7. Transport Study in a BBB Cell Model

  1. Line: 286: I think, it would be good here to very briefly highlight the methodology of how the TEER measurement consisted, what kind of answer will this measurement give? - especially that the following is a description of the permeability experiments and the determination of the permeability coefficient parameter.
  2. Line: 176 (section: 2.3. BBB Cell Model Transport Study Application): please add an explanation of the TEER abbreviation.

  1. Results and Discussion

2.1. Method Development

  1. Line: 84: the explanation of the HBSS abbreviation.

2.2.1. Specificity

  1. Line: 129-131: in the legend of the Figure 2, I propose to put a very brief information on the explanation of the color scheme for individual chromatograms. If it was possible - their better quality = sharpness would be useful in this figure. But this is only a suggestion for possible consideration.

2.3. BBB Cell Model Transport Study Application

  1. Line: 178: “in vitro” in an italic form.

Author Response

Dear professor:

Thank you for your comments concerning our manuscript entitled “Simultaneous determination of seven lipophilic and hydrophilic components in Salvia miltiorrhiza by LC-MS/MS method and its application to a transport study in blood-brain-barrier cell model” (ID: 1535529). Those comments are all valuable and very helpful for revising and improving our paper, and the important guiding significance to our research. We have studied comments carefully and have made corrections which we hope meet with approval. Revised portions are marked in red in the paper. The major modifications in the article and the responses to the comments are as flows:

Introduction

There are some gaps in the text in the space between the name and the abbreviation in square brackets, which require correction in some places.

Response 1: Thank you very much for your careful review. We are very sorry for our incorrect writing, and some spaces were added before the brackets in Line 63-64.

Line: 64-67: in these sentences, I would change the word “anti-apoptosis” on “anti-apoptotic”, and after "enhancement of cholinergic" I would add "action" or "transmission", or something like that.

Response 2: Thank you very much for your careful review. It is really true that some words should be corrected and some words should be added. Line 64-67, the statement of “anti-apoptosis” was corrected as “anti-apoptotic”, the word “transmission” was added after "enhancement of cholinergic".

Line: 75-78: maybe in marking the research goal it is worth mentioning, signaling in what kind of matrix, in what environment such determination was developed? (I am talking about this only because earlier it was mentioned about analytical research in biological material in vivo). But this is only a suggestion for you to consider.

Response 3: Thank you for your careful review. Line 75-78, “in HBSS (Hank's Balanced Salt Solution) samples” was added. In this study, the simultaneous determination of seven components was developed in HBSS samples.

Materials and Methods

3.1. Chemicals

If possible, it would be good to indicate the starting (stock) concentrations of the reagents used or their form (especially that in section 3.4. the authors mention about starting concentrations of CS). But this is only a suggestion to consider.

Response 4: Thank you for your careful review. We have rewritten this part according to the your helpful suggestion. Stock solutions of the TS I, TS IIA, DTS I, and CTS were prepared by dissolving each accurately weighed standard in DMSO and further diluting with methanol to obtain a 10 μg/mL final concentration. Stock solutions of the PAL, PCTA, CFA, and SMZ (IS) were prepared by dissolving each accurately weighed standard in methanol to ob-tain a 1 mg/mL final concentration.

3.3. Instrumentation and Conditions

Line: 224-225: maybe it would be better to put these percentages right next to the numerical values and precede the dash (i.e.: “-“) with a space (something like this: “40% - 50%”)? It seems that the perception of these numerical values would look a bit optically better.

Response 5: Thank you very much for your really careful review. We have rewritten this part as “The mobile phase consisted of 0.05% formic acid in an aqueous solution as solvent A and acetonitrile solution as solvent B. The column was eluted with a gradient of 85% - 50% A at 0-3.5 min, 50% - 10% A at 3.5-6 min, 10% - 5% A at 6-7.5 min, and 5% A at 7.5-14 min.”.

3.4. Stock Solutions, Calibration Solutions and Quality Control Samples

Line: 236: there is no space after the abbreviation SMZ.

Response 6: Thank you very much for your really careful review. Space was added after the abbreviation SMZ.

3.7. Transport Study in a BBB Cell Model

Line: 286: I think, it would be good here to very briefly highlight the methodology of how the TEER measurement consisted, what kind of answer will this measurement give? - especially that the following is a description of the permeability experiments and the determination of the permeability coefficient parameter.

Response 7: Thank you very much for your careful review. We have rewritten this part according to the your helpful suggestion.

The hBMECs cells were seeded at a density of 3×105 cells per well on the Transwell plates in EMC. Apical side volumes were 0.5 mL, and the medium was changed every day. Basal side volumes were 1.5 mL, and the medium was changed every other day. The medium was changed until confluent monolayers were formed. The integrity and transport capacity of the hBMEC cell monolayer was checked by measuring the TEER using an epithelial voltammeter as the protocols suggested by the manufacturer. TEER provides information on ion current resistance across cell monolayers related to the integrity of tight junctions between cells.

Line: 176 (section: 2.3. BBB Cell Model Transport Study Application): please add an explanation of the TEER abbreviation.

Response 8: Thank you very much for your careful review. The explanation of the TEER, trans epithelial electric resistance, was added before “TEER”.

Results and Discussion

2.1. Method Development

Line: 84: the explanation of the HBSS abbreviation.

Response 9: Thank you very much for your careful review. The explanation of the HBSS, Hank's Balanced Salt Solution, was added before “HBSS”.

2.2.1. Specificity

Line: 129-131: in the legend of the Figure 2, I propose to put a very brief information on the explanation of the color scheme for individual chromatograms. If it was possible - their better quality = sharpness would be useful in this figure. But this is only a suggestion for possible consideration.

Response 10: Thank you very much for your careful review. Red, dark green, dark blue, purple, brown, light green, light blue, and yellow represent PCTA, PAL, CFA, SMA, DTS I, CTS, TS I, and TS IIA. The representative meaning of the figure was clearly marked with words in the figure, which is self-evident, so we haven't put it in legend again. We are very sorry for unavailablity of the figures of better quality as they were exported from MassHunter workstation.

2.3. BBB Cell Model Transport Study Application

Line: 178: “in vitro” in an italic form.

Response 11: Thank you very much for your really careful review. We corrected on the form.

We appreciate reviewers’ warm work earnestly and hope that the correction will meet with approval. Once again, thank you very much for your comments and suggestions. If you have any queries, please don’t hesitate to contact me at the address below.

Thank you and best regards.

Yours sincerely,

Zhanying Hong,

hongzhy001@163.com

2022.01.12

Reviewer 2 Report

This study provides sufficient results that display a valuable contribution to the knowledge. There are some minor issues that need to be explained in the manuscript.

*Abbreviations should be written as long in the first place they are used, then the abbreviation should be used. Make any necessary corrections throughout the entire article.

*. It would be better to improve the BBB Cell Model Transport Study Application part, compare and add relevant references. 

*My other suggestions can be seen in the pdf file.

Author Response

Dear professor:

Thank you for your comments concerning our manuscript entitled “Simultaneous determination of seven lipophilic and hydrophilic components in Salvia miltiorrhiza by LC-MS/MS method and its application to a transport study in blood-brain-barrier cell model” (ID: 1535529). Those comments are all valuable and very helpful for revising and improving our paper, and the important guiding significance to our research. We have studied comments carefully and have made corrections which we hope meet with approval. Revised portions are marked in red in the paper. The major modifications in the article and the responses to the comments are as flows:

  1. Abbreviations should be written as long in the first place they are used, then the abbreviation should be used. Make any necessary corrections throughout the entire article.

Response 1: Thank you very much for your careful review. We are very sorry for our incorrect writing, and we have corrected in many places throughout the entire article marked in red in the paper.

  1. It would be better to improve the BBB Cell Model Transport Study Application part, compare and add relevant references.

Response 2: Thank you very much for your careful review. We have rewritten this part and add relevant references according to the your helpful suggestion .Treatment with TS IIA was found to suppress disruption of the BBB, and also polarized transport of CTS was found with facilitated efflux from the abluminal side to luminal side in endothelial cell monolayers, which are consistent with our results and would be scientific evidence to support the application of the components of SM to a certain extent.

  1. Add the author name Bundge.

Response 3: Thank you very much for your careful review. The author name Bundge was added after Salvia miltiorrhiza in the title and the text in 11 places.

  1. write the full version of CNS in the first place and then use the abbreviation.

Response 4: Thank you very much for your careful review. The full version of CNS, central nervous system, was added before “CNS”.

  1. Space

Response 5: Thank you very much for your careful review. We are very sorry for our incorrect writing, and some spaces were added before the brackets in Line 63-64.

We appreciate your warm work earnestly and hope that the correction will meet with approval. Once again, thank you very much for your comments and suggestions. If you have any queries, please don’t hesitate to contact me at the address below.

Thank you and best regards.

Yours sincerely,

Zhanying Hong,

hongzhy001@163.com

2022.01.12